# Endometrial Stromal Senescence Mediates the Progression of Intrauterine Adhesions

**DOI:** 10.3390/ijms26094183

**Published:** 2025-04-28

**Authors:** Pavel I. Deryabin, Aleksandra V. Borodkina

**Affiliations:** Mechanisms of Cellular Senescence Laboratory, Institute of Cytology of the Russian Academy of Sciences, Tikhoretsky Ave. 4, Saint-Petersburg 194064, Russia; deryabin.pav@gmail.com

**Keywords:** endometrium, senescence, intrauterine adhesions, Asherman’s syndrome, LGALS9

## Abstract

Cellular senescence has emerged as a key mediator in organ-specific fibrosis. Here, we have established the role of endometrial stromal senescence in the progression of endometrial fibrosis, termed intrauterine adhesions (IUA). IUA have significant negative effects on women’s reproductive health and are associated with infertility. We have generated original gene signatures to identify endometrial stromal senescence in single-cell and bulk RNA-sequencing data. By applying generated gene signatures, we revealed an increased level of stromal senescence during the proliferative phase in the endometrium of patients with IUA. Further comparative analysis of cell–cell communications demonstrated that senescent stromal cells in the IUA endometrium create an immunosuppressive and profibrotic microenvironment through an elevated expression of *LGALS9*. Endometrial stromal senescence persists during the window of implantation and correlates with impaired embryo receptivity of the IUA endometrium. Therefore, stromal senescence can be regarded as a primary cause of an unresponsive endometrium with decreased receptivity and thickness in IUA patients. A LGALS9 immunotherapy protocol, specifically designed to neutralize LGALS9 immunosuppressive activity of senescent cells, may offer a promising opportunity to restore effective immune clearance of these cells within the IUA stroma. Consequently, an LGALS9-based strategy could emerge as a novel therapeutic avenue in the treatment of IUA.

## 1. Introduction

Intrauterine adhesions (IUA), also known as Asherman’s syndrome, are characterized by endometrial fibrosis, which leads to the partial or complete obliteration of the uterine cavity due to adhesions of the uterine wall [1]. The main cause of IUA is trauma to the endometrial basal layer of the endometrium, mainly post-dilation and curettage [2]. However, this condition can also occur after hysteroscopic surgical procedures, including hysteroscopic myomectomy and hysteroscopic adenomyomectomy, which breach the basal layer of the endometrium or the myometrium [3]. Common clinical manifestations of IUA are menstrual abnormalities, pelvic pain, infertility, recurrent miscarriages, and abnormal placentation [1]. The prognosis and treatment outcomes for IUA patients are closely related to the severity of the disease [4]. Therefore, novel classification systems for IUAs are constantly being developed, including those based on a comprehensive analysis of ultrasonographic endometrial thickness, menstrual patterns, reproductive history, and hysteroscopic findings [3,4,5]. Currently, hysteroscopic adhesiolysis, postoperative placement of intrauterine contraceptive devices, low-dose aspirin, and regular estrogen therapy are employed as clinical treatments for IUA [1,6]. These standard approaches provide benefits for patients but exhibit a high recurrence rate of endometrial fibrosis [6]. Consequently, numerous novel therapies targeting the pathophysiological mechanisms of IUA are currently under investigation [1]. While some of these emerging approaches show promising results, most—including stem cell therapy (utilizing intact, genetically modified, or preconditioned stem cells) and direct administration of cytokines—address the consequences rather than the underlying causes of the disease. Therefore, a comprehensive investigation of the pathological changes in the endometrium of IUA patients is essential to uncover the molecular mechanisms underlying this condition, which may serve as primary therapeutic targets for effectively treating IUA.

It has been reported that the pathological process of tissue fibrosis is associated with the accumulation of senescent cells [7]. Senescence is a common stress response characterized by irreversible cell cycle arrest, impairments in various intracellular organelles and systems, and the secretion of numerous pro-inflammatory and bioactive molecules collectively referred to as the senescence-associated secretory phenotype (SASP) [8]. Current research indicates that senescent cells directly contribute to the development of fibrotic diseases through profibrogenic factors present in the SASP, notably transforming growth factor beta (TGF-β), interleukin-11 (IL-11), and SERPINE1 [9,10,11]. Furthermore, the elimination of senescent cells using genetic approaches or senolytics in both mouse and human models of organ-specific fibrosis leads to significant improvements in lung, kidney, and cardiac function [9,12,13,14].

Being the primary structural and functional component of the endometrium, endometrial stromal cells (EnSC) are capable of scarless tissue regeneration, the formation of the functional layer through active estrogen-driven proliferation, and the establishment of decidual tissue via progesterone-induced differentiation [15,16]. Recent data highlight the role of EnSC in the development of IUA, although the precise pathogenesis remains to be elucidated [17]. Indeed, EnSC isolated from patients with IUA exhibit reduced colony-forming, migration, invasion, angiogenic, and immunosuppressive abilities compared to those from healthy women [18]. We have previously demonstrated that senescence is a primary response of EnSC to various stressful insults [19,20]. EnSC display all the classical features during the development of senescence, including SASP secretion [21,22,23]. In terms of functionality, senescent EnSC become refractory to sex steroid hormones, rendering them unable to proliferate and decidualize properly [24]. By producing SASP, senescent EnSC create an unfavorable microenvironment for embryo implantation [24,25]. Considering the above, we propose that the accumulation of senescent stromal cells may contribute to the progression of IUA.

## 2. Results

### 2.1. Endometrium-Specific Gene Signatures for Identifying Stromal Senescence in Transcriptomic Data

To identify senescent stromal cells in endometrial samples from patients with IUA, we employed three distinct approaches, each based on independent senescence-related genesets (Appendix A). The first geneset, “Fridman senescence up,” comprises 77 genes primarily associated with cell cycle arrest [26]. The second geneset, “Saul SenMayo”, consists of 125 genes that mainly represent SASP factors [27]. Although both senescence genesets are commonly used to identify senescent cells in transcriptomic data, they lack cell-type specificity, as senescent cells are highly heterogeneous, and their transcriptomic profiles largely depend on their cellular origin. For the purpose of this study, we generated original senescence genesets that are more specific to endometrial tissue. To achieve this, we conducted differential expression analysis (DEA) between cultured undifferentiated control and senescent EnSC using our bulk RNA-sequencing (RNA-seq) data (GSE160702) (Figure 1A).

Based on the DEA results, we created two lists of coordinated gene expressions, as follows: the first comprised the top 100 up-regulated differentially expressed genes (DEGs) (Top 100 up-regulated DEGs) and the second included the top 100 down-regulated genes (Top 100 down-regulated DEGs). As shown in Figure 1B, all tested genesets exhibited minimal overlap (Figure 1B). Concurrently, further functional enrichment analysis revealed several biological processes typical for senescence that were overrepresented in each geneset (Figure 1C, Appendix A). As anticipated, classical senescence genesets were enriched with terms related to proliferation arrest and SASP production (Figure 1C, Appendix A). Our “Top 100 down-regulated DEGs” signature also included biological processes associated with cell cycle arrest, whereas the “Top 100 up-regulated DEGs” were more closely linked to SASP (Figure 1C, Appendix A). In summary, we generated two novel genesets that can be further utilized to identify senescence in the endometrial stroma during the proliferative phase of the menstrual cycle.

### 2.2. Enhanced Senescence in the Endometrial Stroma of Patients with IUA During the Proliferative Phase

To assess the level of senescence in the endometrium of patients with IUA, we first analyzed relevant single-cell RNA-sequencing (scRNA-seq) datasets PRJNA730360 and PRJNA788201. These datasets included normal endometrial samples and samples from patients with IUA obtained during the late proliferative phase. Initially, we integrated the datasets, obtained and annotated clusters as indicated in Figure 2A–C, and selected a subset of stromal cells for further analysis.

Next, we compared the levels of senescence in healthy stroma versus stroma from patients with IUA by calculating single-cell enrichment scores for each geneset. As shown in Figure 2D, both the “Fridman senescence up” and “Saul SenMayo” genesets demonstrated higher enrichment scores in the IUA stroma, indicating an increased level of senescence in the endometrium of IUA patients (Figure 2D). Moreover, the results obtained using genesets of in vitro EnSC senescence coincided with those for classical senescence genesets. Specifically, the “Top 100 up-regulated DEGs” signature was higher in IUA stroma, whereas the “Top 100 down-regulated DEGs” was lower (Figure 2D). Additionally, to demonstrate the specificity of the generated genesets that mark senescent EnSC, we assessed the level of senescence in thin endometrium alongside the IUA samples. Previous data provided evidence for increased senescence in the stroma of thin endometrium [28]. Indeed, the scores obtained for the “Top 100 up-regulated DEGs” and the “Top 100 down-regulated DEGs” gene signatures confirmed enhanced senescence in thin endometrial stroma and exhibited a similar trend to those observed in IUA stroma (Appendix A). To further validate our findings, we performed an additional analysis. The results of this analysis are presented in Appendix A. To begin with, we selected all cells of mesenchymal origin, including stromal cells, myofibroblasts, and proliferating stromal cells and reintegrated the data to identify any new stromal subclusters that might emerged in IUA and to reveal potential subclusters of senescent cells (Appendix A). As shown in Appendix A, no cluster specific to IUA was identified (Appendix A) and no distinct senescent cell cluster was observed (Appendix A). Notably, to estimate senescence in all mesenchymal cell types, we applied the “Saul SenMayo” geneset, as our newly generated signatures are specific to stromal cells. Senescence signal was uniformly distributed throughout the entire cell population within each subtype. However, a pairwise comparison showed that the senescence signal among mesenchymal cell subtypes primarily developed in the stroma (Appendix A). The results of this analysis justify the relevancy of studying stromal cell senescence in IUA pathogenesis. Collectively, our findings revealed novel genesets for identifying senescence in endometrial stroma, based on knowledge transferred from the in vitro EnSC senescence program to in vivo tissue samples. To further validate the elevated senescence in the endometrial stroma of patients with IUA, we utilized two available bulk RNA-seq datasets (GSE224093 and PRJNA916532) that included samples of normal and IUA endometrium during the late proliferative phase. Before analyzing the bulk samples, we employed scRNA-seq data to assess the relative expression of each gene signature across various endometrial cell types. As shown in Figure 2E, the “Fridman senescence up” geneset was found to be more specific to myofibroblasts, while the “Saul SenMayo” geneset demonstrated comparable specificity across most cell types in the endometrium (Figure 2E). Consequently, both genesets are unsuitable for investigating stromal senescence in bulk data. In contrast, the “Top 100 up-regulated DEGs” and the “Top 100 down-regulated DEGs” gene signatures exhibited greater specificity for stromal cells compared to other cell types in the endometrium, with the exception of myeloid cells (Figure 2E). To account for the cell composition in the analyzed bulk samples, we applied deconvolution and revealed a significantly higher stromal content in all endometrial samples, while myeloid cells constituted only about 2–3% (Figure 2F). This suggests a minimal, if any, impact of myeloid cells on the assessment of senescence. Consistent with the results obtained from scRNA-seq data, we observed significant upregulation of the Top-up genes and downregulation of the Top-down genes in IUA compared to normal endometrium (Figure 2G). These findings support the conclusion of increased stromal senescence in the IUA endometrium.

### 2.3. Senescent Stromal Cells in the IUA Endometrium Create an Immunosuppressive and Profibrotic Microenvironment Through Increased Expression of LGALS9

To gain further insights into the biological roles of stromal senescence in the IUA endometrium, we conducted a comparative analysis of cell–cell communications. As shown in Figure 3A, the number and strength of interactions, both incoming and outgoing from stromal cells, significantly increase in the endometrium of IUA patients (Figure 3A). We then aimed to delineate the signaling alterations in the IUA stroma. To achieve this, we identified the up-regulated ligand–receptor pairs signaling from stromal cells in the IUA endometrium compared to normal tissue (Figure 3B, Appendix A). As expected, EnSC from patients with IUA activated several canonical SASP pathways, including CXCL, VEGF, IGF, IFGBP, and PLAU (Figure 3B). Furthermore, IUA stromal cells exhibited elevated levels of MDK, thereby activating related signaling in receptor-producing counterparts. Notably, a recent study has identified midkine (encoded by MDK) as an age-related biomarker that mediates aging-related phenotypes [29].

Finally, we identified active LGALS9 signaling specifically present in the stromal cells of patients with IUA (Figure 3B). Galectin-9, encoded by the *LGALS9* gene, is currently recognized as an immune system inhibitor [30]. The identified ligand–receptor pairs involving LGALS9 included LGALS9-CD45, LGALS9-CD44, LGALS9-HAVCR2, and LGALS9-P4HB (Figure 3B). As expected, increased expression of three LGALS9 receptors—CD44, CD45, and HAVCR2—was observed in myeloid and lymphoid cells from the IUA endometrium. This finding provides evidence for the inhibitory effect of stromal cells on immune cells during IUA progression. The expression of another receptor for LGALS9, P4HB, was enhanced in the myofibroblasts, proliferating stromal cells in IUA endometrium (Figure 3B). Since P4HB is the primary collagen cross-linking enzyme involved in collagen biosynthesis, the LGALS9–P4HB interaction reflects the profibrotic action of stromal cells during IUA progression. To visualize the differences in LGALS9 signaling between normal and IUA stroma, we compared the aggregated Galectin signaling networks under both conditions. As shown in Figure 3C, no LGALS9 signals were outgoing from the stromal cells in the normal endometrium. In contrast, stromal cells from the IUA endometrium displayed outgoing LGALS9 signals to nearly all cell types in the tissue (Figure 3C). Importantly, we found a strong correlation between *LGALS9* expression in stromal cells and cellular senescence estimated using “Top 100 up-regulated DEGs” enrichment scores (Figure 3D). Overall, the results indicate that the endometrial stroma of IUA patients is characterized by elevated senescence and exhibits immunosuppressive and profibrotic actions, at least in part due to increased expression of *LGALS9*. The LGALS9-mediated inhibitory effect of stromal cells on immune cells may hinder the clearance of senescent EnSC, allowing them to persist in the endometrium of IUA patients.

### 2.4. Endometrial Stromal Senescence Persists During the Window of Implantation (WOI) and Is Associated with Impaired Decidualization of the IUA Endometrium

Finally, we investigated whether senescence detected in the endometrial stroma of patients with IUA during the proliferative phase can be transmitted to the secretory phase, specifically to WOI. Firstly, we generated novel genesets suitable for identifying senescence in the endometrial stroma upon decidualization. Following the approach described above, we conducted DEA between control and senescent EnSC on the fourth day of decidualization (reflecting WOI) using our bulk RNA-seq data (GSE160702) and created “Top 100 up-regulated DEGs” and “Top 100 up-regulated DEGs” lists with coordinated gene expressions (Figure 4A).

The purpose of the obtained genesets was to differentiate between normal and senescent EnSC during decidualization. As anticipated, both genesets developed for decidualized EnSC showed minimal overlap with those for undifferentiated cells (Figure 4B). Consistent with this, the new “Top 100 down-regulated DEGs” list comprised genes related to the differentiated decidual EnSC state, whereas the new “Top 100 up-regulated DEGs” list included genes related to the undifferentiated and inflammatory states (Figure 4C, Appendix A).

To assess the level of stromal senescence during the WOI, we proceeded with the analysis of the scRNA-seq dataset (GSE215968), which included relevant control and IUA samples (Figure 4D). As illustrated in Figure 4E, calculated enrichment scores for both the “Top 100 up-regulated DEGs” and “Top 100 down-regulated DEGs” gene signatures indicated that the higher level of senescence in the endometrial stroma from patients with IUA, observed during the proliferative phase, persisted also during the WOI (Figure 4E). According to our earlier in vitro data, senescent EnSC are refractory to decidual stimuli and, therefore, unable to differentiate into mature and senescent decidual cells [24]. The impaired emergence of these decidual subpopulations is believed to contribute to reproductive failure [31,32,33]. Given the increased senescence in the IUA stroma, we hypothesized that this condition might lead to impaired formation of decidual subpopulations during the WOI. To test this hypothesis, we assessed enrichment scores by applying previously generated gene signatures for pre-decidual, mature and senescent decidual cell [31]. As suggested, the IUA stroma exhibited a reduced emergence of each subpopulation (Figure 4F). Therefore, the functional outcome of heightened stromal senescence during the WOI appears to be the impaired generation of decidual subpopulations, which may contribute to the infertility observed in patients with IUA.

## 3. Discussion

Cellular senescence has emerged as a key mediator in organ-specific fibrosis, including idiopathic pulmonary fibrosis, acute kidney injury, chronic kidney disease, etc. [7,9,10,12,13,14,34]. For instance, biospecimens from the lungs of patients with idiopathic pulmonary fibrosis exhibited elevated levels of senescence biomarkers p16 and γH2A.X compared to those from healthy controls [9]. Furthermore, senescent cells have been identified in various tissue compartments in chronic kidney disease, including the proximal tubular epithelium, glomeruli, and endothelium [35]. In the context of cardiac fibrosis, the accumulation and persistence of senescent cardiac fibroblasts contribute to chronic inflammation, collagen deposition, myocardial hypertrophy, and the progression of fibrosis [36]. Current knowledge indicates that the profibrotic effects of senescent cells are primarily mediated by SASP, which comprises pro-inflammatory cytokines such as IL-6, IL-8, CXCL8, CXCL12, TNF-α, MCP-1, and MIP-1; reactive oxygen species; matrix metalloproteases; extracellular matrix proteins; and remodelers [7,9,10,12,13,14,34,35,36]. In this study, we investigated whether enhanced stromal senescence contributes to the development and progression of another fibrotic condition—IUA. The only reference to the role of senescence in IUA progression coincided with our investigation [37]. The authors have highlighted the crucial role of endothelial cell senescence in IUA pathogenesis, contributing to vascular reduction and fibrosis. Notably, the immunofluorescent co-staining of p21/p16 (senescent cell markers) and CD31 (endothelial cell marker) conducted in this study provides experimental evidence that senescence in the IUA endometrium is not confined to endothelial cells, but is likely also present in stromal cells. Although the authors did not address this finding, it underscores the critical importance of studying stromal senescence in IUA. Furthermore, it supports our hypothesis regarding the potential role of senescent stromal cells in the pathogenesis of IUA.

One of the main challenges in studying senescence is its heterogeneity, which underscores the need for improved technologies to accurately identify and characterize senescent cells. At present, numerous genesets and approaches have been described to identify senescent cells in transcriptomic data, including SenMayo, SnG signatures, SenePy, and SenCid [27,38,39,40,41]. Unfortunately, none of these approaches appear to be universally applicable. We believe that the best way to study the senescence of the desired cell type within tissues is to generate cell-type-specific senescence signatures, instead of using universal SnG signatures or machine learning models. In the present study, we developed an original approach to identify senescence in the endometrial stroma, drawing on knowledge learned from the in vitro EnSC senescence program and transferring this knowledge to in vivo tissue samples. The accuracy of our approach was validated by comparing it with two of the most commonly used senescence genesets and by reproducing results previously obtained for thin endometrium [28]. Having established the validity of our approach for assessing endometrial senescence, we have discovered enhanced senescence in the endometrial stroma of patients with IUA using five independent RNA-seq datasets.

The potential consequences of elevated senescence on tissue functioning arise directly from the primary characteristics of senescent cells, as follows: irreversible proliferation arrest and altered auto-paracrine activity. Regarding the former, the increased senescence observed in the endometrial stroma of patients with IUA during the proliferative phase may adversely affect EnSC proliferation and the proper formation of a new functional layer. Consistent with this, the endometrium in women with IUA is significantly thinner than that of women without IUA, and EnSC isolated from IUA patients demonstrate a reduced colony-forming ability compared to those from healthy women [18,42,43]. To address the functional outcomes of the altered secretory activity of senescent cells, we compared the cell–cell communications between normal and IUA stroma. Our findings revealed that stromal cells from patients with IUA are “much more sociable” than those from normal tissue, interacting with a broader range of cell types in the endometrium. Firstly, we detected that EnSC from IUA patients displayed enhanced expression of several canonical SASP components. We have previously shown that senescent EnSC via SASP can induce paracrine senescence in normal bystander cells, thereby facilitating the negative effects of senescence [22]. Furthermore, such inflammatory molecules produced by senescent EnSC, such as CXCL8 and CXCL12, may be a source of chronic inflammation in the IUA endometrium. Additionally, we revealed that senescent stromal cells in the IUA endometrium are characterized by enhanced expression of the urokinase-type plasminogen activator (PLAU), which activates the urokinase-type plasminogen receptor (PLAUR) in myeloid, endothelial, epithelial, stromal, and myofibroblast cells. PLAUR is now recognized as a central immune-regulating receptor, which can alter intracellular signaling, cell activation, extracellular adhesion, and migration [44]. In addition to classical SASP components, stromal cells from IUA patients exhibited a marked increase in *LGALS9* expression, thereby creating a pro-inflammatory, immunosuppressive, and profibrotic microenvironment. In accordance with our results, a recent study described a novel immunosuppressive subtype of senescent tumor cells that exerted an inhibitory effect on immune cells, including mast cells, myeloid cells, NK cells, B cells, and T cells via LGALS9 signaling [45]. Currently, LGALS9 has emerged as a promising new target for cancer immunotherapy [46]. Beyond cancer, it has been observed that LGALS9 plays a role in bone-marrow-derived mesenchymal stem-cell-mediated immunosuppression [47]. This occurs by LGALS9 binding to its receptor, HAVCR2, expressed on activated lymphocytes, leading to apoptotic cell death of these activated lymphocytes [47]. Furthermore, the immunosuppressive action of LGALS9 has also been demonstrated in relation to reproductive tissues [48]. Research has shown that the LGALS9/HAVCR2 mechanism is crucial for establishing and maintaining maternal–fetal immune tolerance, which is necessary for the maternal immune system to tolerate the semi-allogeneic fetus. Specifically, the authors revealed that trophoblasts induced the transformation of peripheral blood natural killer (NK) cells into more immune-tolerant uterine NK (uNK) cells (positive for HAVCR2) through the secretion of LGALS9 and the interaction between LGALS9 and HAVCR2. Additionally, it has been established that EnSC also secrete LGALS9, which may interact with HAVCR2 expressed on a subset of uNK cells [49]. Traditionally, senescent cells, including those in the endometrial stroma, are eliminated from tissues by NK/uNK cells [50,51]. Therefore, senescent stromal cells in the endometrium of patients with IUA during the proliferative phase may hinder immune clearance by activating LGALS9 signaling pathways in uNK cells. Indeed, by employing our original approach and generating additional senescence signatures suitable for the decidualized EnSC state, we observed that elevated senescence in the IUA stroma persisted from the proliferative to the mid-secretory phase. In our previous study, we conducted a detailed investigation into how senescent EnSC impair embryo implantation [24]. According to our previous in vitro findings, senescent EnSC are refractory to decidual stimuli and are unable to differentiate into either mature or senescent decidual cells [24]. The reduced emergence of decidual subpopulations is thought to contribute to embryo implantation failure [31,32,33]. Consistent with our in vitro findings, we have demonstrated that heightened senescence in the IUA stroma during the WOI resulted in a decreased emergence of pre-decidual cells, as well as both mature and senescent decidual subpopulations that are required for proper embryo implantation. Interestingly, IUA is typically characterized by a hormonally unresponsive endometrium that is unable to support proper embryo implantation [52]. In cycles without conception, heightened stromal senescence may affect “scarless” healing of the endometrium. Although certain amounts of senescent cells are essential for wound healing by favoring the plasticity of different cell populations necessary for wound closure, incomplete elimination and accumulation of these senescent cells could promote impaired tissue formation due to the deposition of collagen, leading to fibrotic tissue [53].

In conclusion, we have identified stromal senescence as the underlying cause of unresponsive endometrium, which is characterized by decreased receptivity and thickness in patients with IUA. The presence of senescent cells may adversely affect or diminish the efficacy of existing treatments for IUA, potentially leading to the recurrence of fibrosis after hysteroscopic adhesiolysis, insensitivity of the endometrial stroma to estrogen stimulation, or paracrine senescence of transplanted stem cells, thereby compromising their functionality. We propose that targeting senescent stromal cells could represent a novel therapeutic approach in the treatment of IUA. We believe that the most effective way to minimize the complex profibrotic and pro-inflammatory effects of senescent cells is through their elimination. In this context, focusing on the specific SASP factors does not appear to be a promising strategy for reducing the overall impact of SASP. Moreover, given that traditional senolytic strategies often have side effects due to imperfect selectivity, a more promising alternative for curing IUA may be an LGALS9 immunotherapy protocol, specifically designed to neutralize LGALS9 immunosuppressive activity of senescent cells. This approach may facilitate the restoration of proper immune clearance of senescent cells in the IUA stroma. Either alone or in combination with existing treatment modalities, an LGALS9-based strategy has the potential to effectively address severe IUA. At the same time, according to the most recent data, LGALS9 expression in stromal cells is closely linked to their degree of decidualization [54]. Therefore, the potential application of LGALS9 immunotherapy for IUA curing should be strictly limited to proliferative phase of the menstrual cycle, in order to avoid any undesirable impact on decidualization.

The results of the present study are based solely on bioinformatic analysis; therefore, several important limitations should be highlighted. Firstly, the sample size of the publicly available scRNA-seq datasets analyzed in this study is relatively small, indicating a clear need for future research to expand the sample size for further validation. Secondly, this study lacks experimental validation; therefore, the data obtained should be interpreted with caution until experimental validation is complete. Experimental validation requires the isolation of numerous endometrial stromal cell lines from patients with IUA to confirm higher level of senescence and elevated *LGALS9* expression compared to stromal cells obtained from healthy donors. However, there are significant limitations regarding IUA patients, as follows: (1) patients of reproductive age with severe IUA are relatively rare; and (2) most patients with severe IUA possess only small fragments of undamaged endometrium not obliterated by fibrotic adhesive bands. Consequently, to preserve the potential for reproductive success, clinicians often cannot collect the amount of tissue necessary for isolating endometrial stromal cells without risking harm to the remaining endometrium. We have already begun isolating endometrial stromal cell lines from IUA patients, so we hope to validate the results of our in silico analyses in the near future. Meanwhile, our study is the first to highlight the impact of endometrial stromal senescence in IUA pathogenesis. This finding opens up new opportunities to treat IUA, which may include not only the proposed LGALS9 immunotherapy, but also more traditional approaches to overcome senescence, such as the application of senomorphics or senolytics.

## 4. Materials and Methods

### 4.1. Data Collection

This study was based on publicly available human RNA-seq datasets deposited in the Gene Expression Omnibus (GEO) database. To identify senescence-related changes in gene expression specific to EnSC, bulk RNA-seq dataset GSE160702 of the in vitro EnSC senescence model was used. Samples for normal proliferating EnSC and stress-induced prematurely senescent EnSC without decidualization stimulation (Day 0 samples) and under stimulation of decidualization (Day 4 samples) were selected for the analysis [20]. To study changes in gene expression in the endometrial tissue samples associated with IUA, the database was screened with the terms ‘Intrauterine adhesion’, ‘IUA’, and ‘Asherman’. The following five datasets were selected: GSE224093 and PRJNA916532 bulk RNA-seq datasets; and PRJNA730360, PRJNA784021, and GSE215968 scRNA-seq datasets. Sample ‘Control5’ (SAMN32479471) in dataset PRJNA916532 was excluded from the analysis due to an initial RNA quality issue, as suggested by the authors of the original study [55]. scRNA-seq dataset GSE215968 was analyzed using samples from the original study [56].

### 4.2. Data Processing

Conventional command-line and R (v. 4.4.2) packages were used. All datasets, except for the scRNA-seq GSE215968, were downloaded from the GEO in raw format using sra-tools (v. 3.2.0). The raw datasets underwent transcript quantification using salmon (v. 1.9.0) quant and alevin tools, with a reference index built on GENCODE Human Release 41 (GRCh37), as previously described [24]. Alevin has been run with additional parameter—exspectCells set to 10,000 cells.

For the bulk RNA-seq datasets, transcript quantification matrices were imported into the R environment and summarized to a gene level using tximeta (v. 1.24.0). Metadata for the datasets were retrieved using GEOquery (v. 2.74.0). Library normalization was performed using DESeq2 (v. 1.46.0).

For scRNA datasets PRJNA730360 and PRJNA784021, calculated gene expression matrices were imported into R using tximport (v. 1.34.0) and fishpond (v. 2.12.0). Basic data filtering was performed to retain cells with more than 300 and less than 5000 features identified and less than 5% of reads originating from mitochondrial transcripts. Subsequently, DoubletFinder (v. 2.0.4) was applied to remove potential cells aggregates, and the algorithm was applied at 5% doublets rate. Following the initial filtering, the standard seurat (v. 5.2.1) pipeline was employed to process the data, and the standard log-normalization procedure was used. Harmony (v. 1.2.3) was applied to account for donor-specific variability. The obtained cell clusters were annotated using the cell-type marker genes described by the authors of the original studies [28,57]. The scRNA-seq dataset GSE215968 was downloaded in a filtered and annotated quantified format (sc_AS_vs_WOI_Control.h5ad) and processed using seurat, as described above.

### 4.3. Data Analysis

Genesets “Fridman senescence up” and “Saul SenMayo” were retrieved from the Molecular signatures database (MSigDB) under accession numbers M9143 and M45803, respectively. Genesets ‘SS3 PreDC,’ ‘SS4 MatDC,’ and ‘SS5 SenDC,’ which mark subpopulations of decidualizing EnSC, were extracted from study [31]. To generate new genesets specific to EnSC senescence, standard DESeq2 pairwise Wald testing with the Benjamini–Hochberg (BH) *p*-value correction procedure was applied to estimate the DEGs between the senescent and control EnSC from bulk RNA-seq dataset GSE160702, and top DEGs were selected by Log2 fold change among genes estimated at p.adj < 0.1. Functional annotation of genelists was performed using Gene Ontology Biological process terms through unweighted overrepresentation analysis using msigdbr (v. 7.5.1) and clusterProfiler (v. 4.14.4) based on hypergeometric distribution evaluation. Weighted single-cell geneset enrichment analysis was conducted using the rank-based Ucell algorithm (v. 2.11.1). Weighted geneset enrichment analysis for DEA results obtained from bulk RNA-seq datasets was performed using the GSEA algorithm implemented in the clusterProfiler package. Deconvolution of the bulk RNA-seq samples was performed using the GEDIT (v. 3.0) package, input gene expression matrices were variance stabilized and transformed using DESeq2, and the reference matrix was constructed based on log-normalized data for integrated PRJNA730360 and PRJNA784021 analysis objects aggregated by identified cell types. The analysis of cell–cell communication was conducted using CellChat (v. 2.1.2) following a standard comparative analysis pipeline, with differential signaling being estimated by the DEA pipeline. To reduce the droplet effect and impute scRNA-seq data, the ALRA algorithm from the SeuratWrappers (v. 0.4.0) package was utilized, and calculations were performed based on the top 2000 highly variable genes. To evaluate the significance of the difference between the two groups, the Wilcox rank sum test was applied, and the BH procedure was used for multiple testing correction. To assess the correlation between cell features in the scRNA-seq data, the Spearman method was used.

## Figures and Tables

**Figure 1 ijms-26-04183-f001:**
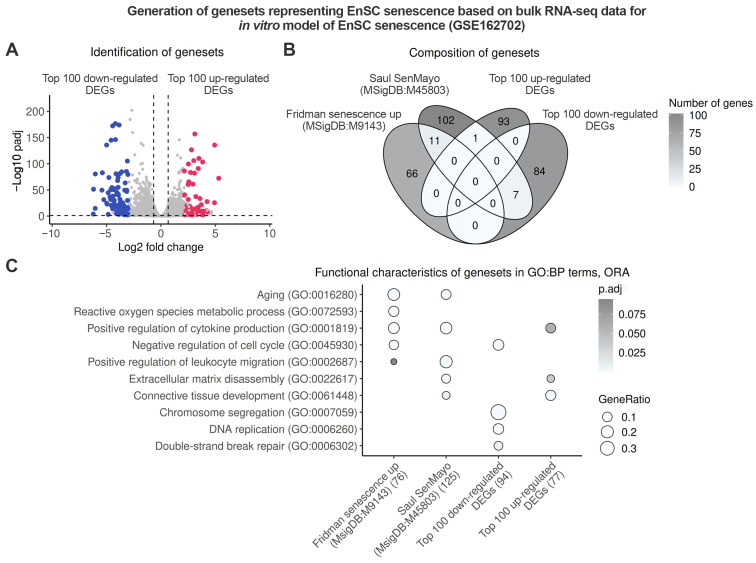
Identification of genesets to detect stromal senescence in proliferative endometrium. Bulk RNA-seq data GSE162702 were utilized to estimate DEGs between senescent and normal proliferating EnSC in vitro. (**A**) Volcano plot displaying selected DEGs. (**B**) Venn diagram illustrating the intersections of identified genesets with classic senescence genesets of ‘Fridman senescence up’ and ‘Saul SenMayo’. (**C**) Evaluation of biological processes mediated by the analyzed genesets.

**Figure 2 ijms-26-04183-f002:**
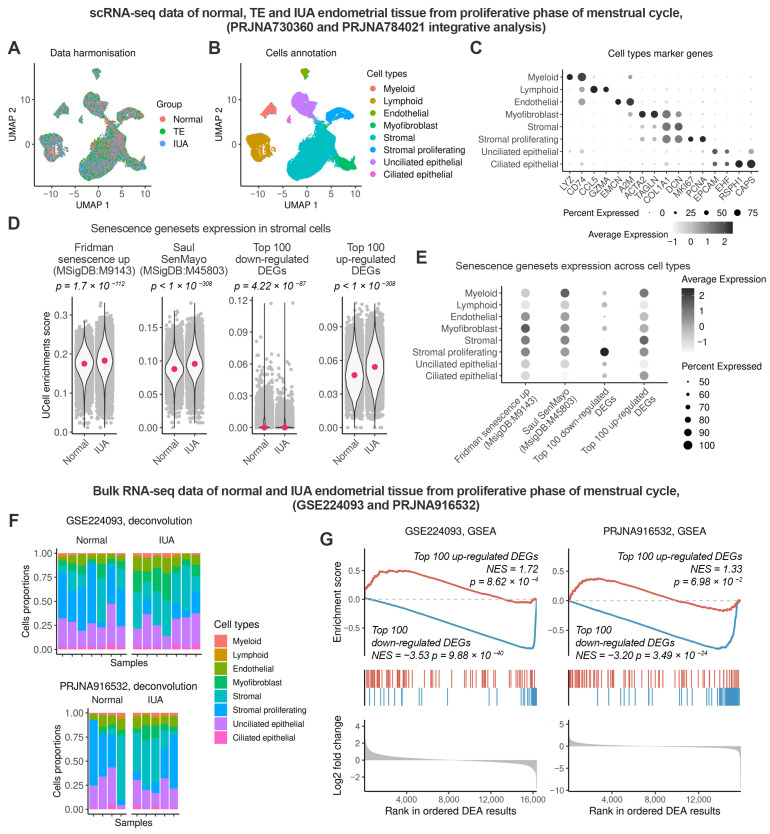
The endometrium of patients with IUA is characterized by an increased level of stromal senescence during the proliferative phase compared to healthy donors. The expression of classic and EnSC-specific senescence gene signatures was assessed in endometrial tissue at the single-cell and bulk tissue sample levels. Normal—healthy donors, TE—patients with thin endometrium, IUA—patients with intrauterine adhesions. Single-cell analysis was conducted by integrating two scRNA-seq datasets, PRJNA730360 (Normal *n* = 3, TE *n* = 3) and PRJNA784021 (IUA *n* = 3). (**A**) UMAP embedding of integrated scRNA-seq data colored by the donors group. (**B**) UMAP embedding of integrated scRNA-seq data colored by annotated cell types. (**C**) The expression of commonly used marker genes of identified cell clusters. (**D**) Comparison of the expression of classic and EnSC-specific senescence genesets in stromal cells between Normal donors and IUA patients. (**E**) The cell-type specificity of classic and EnSC-specific senescence genesets. Evaluation of stromal senescence at the bulk tissue sample level was performed using EnSC-specific senescence genesets, and the GSE224093 (Normal *n* = 7, IUA *n* = 7) and PRJNA916532 (Normal *n* = 4, IUA *n* = 5) datasets were analyzed. (**F**) The cell composition of cell types in bulk RNA-seq samples. (**G**) GSEA enrichment analysis of the genesets across a ranged list of DEGs between IUA and Normal samples.

**Figure 3 ijms-26-04183-f003:**
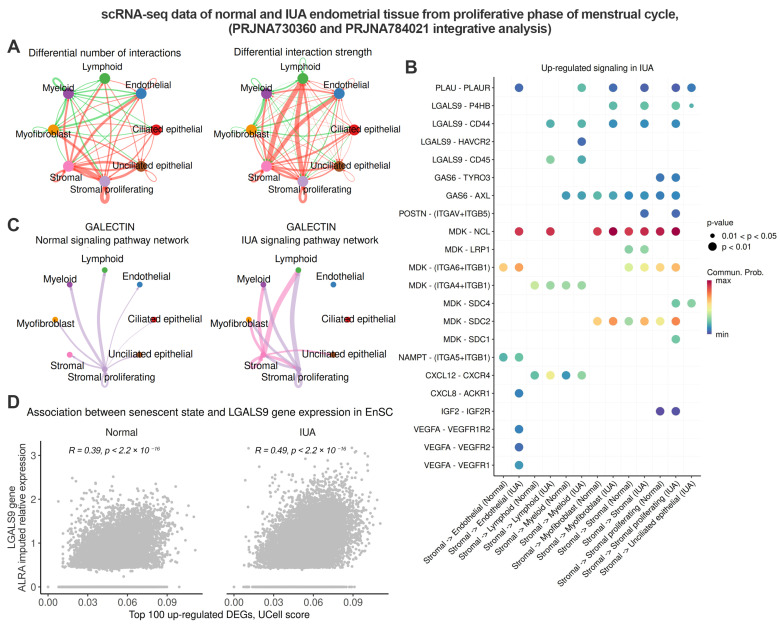
Stromal cells of patients with IUA exhibit immunosuppressive and profibrotic secretory activity. Normal—healthy donors, IUA—patients with intrauterine adhesions. The analysis was performed on scRNA-seq datasets PRJNA730360 (Normal *n* = 3) and PRJNA784021 (IUA *n* = 3), integrated, and annotated, as demonstrated in Figure 2. (**A**) Circle plots depict the differential number of interactions and the strength of interactions among identified cell populations between IUA and Normal groups, with orange/green edges representing increased/decreased signaling in IUA compared to the Normal group. (**B**) Up-regulated in IUA secreted signaling outgoing from stromal cells. (**C**) Circle plot showing interaction strength of Galectin signaling outgoing from stromal cells in Normal and IUA groups. (**D**) Associations between LGALS9 gene expression and ‘Top 100 up-regulated DEGs’ geneset enrichment scores in stromal cells in Normal and IUA groups.

**Figure 4 ijms-26-04183-f004:**
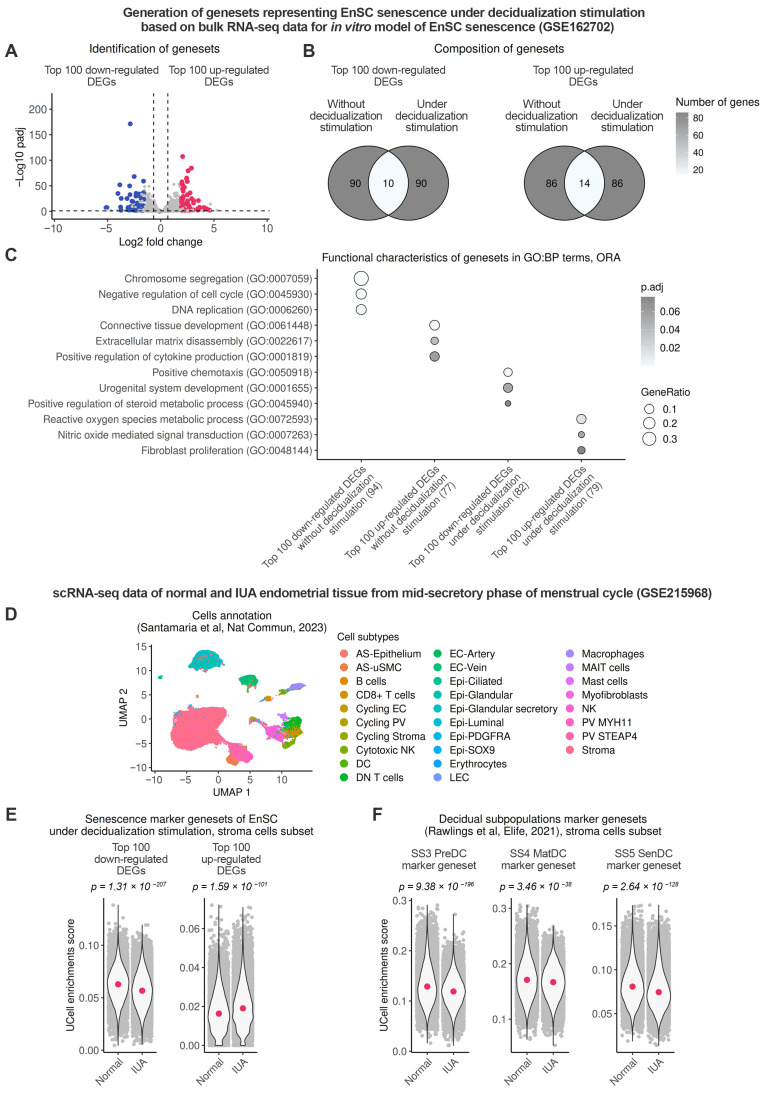
Stromal cells from patients with IUA retain the senescent phenotype and lose the ability to decid-ualize during the mid-secretory phase compared to healthy donors. Normal—healthy donors, IUA—patients with intrauterine adhesions. To differentiate between normal and senescent EnSC during decidualization, bulk RNA-seq data GSE162702 were used to estimate DEGs between se-nescent and normal EnSC exposed to decidualization stimulation in vitro. (**A**) Volcano plot dis-playing selected DEGs. (**B**) Venn diagram illustrating the intersections of identified genesets with genesets characterizing EnSC senescence in an undifferentiated state. (**C**) An evaluation of the bi-ological processes mediated by the analyzed genesets. To compare the endometrium from the pa-tients with IUA and normal endometrium during the mid-secretory phase, scRNA-seq datasets GSE215958 (Normal *n* = 6, IUA *n* = 9) were investigated. (**D**) UMAP embedding of integrated scRNA-seq data colored by cell types. (**E**,**F**) A comparison of expression identified EnSC-specific senescence genesets and decidual subpopulation marker genesets in stromal cells between Normal donors and IUA patients [31].

## Data Availability

All data used in this study are available in the GEO database under the indicated GEO accession numbers.

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
