# Peer review of "Endometrial Stromal Senescence Mediates the Progression of Intrauterine Adhesions"

_ijms, 2025, doi:10.3390/ijms26094183_

Round 1

Reviewer 1 Report

Comments and Suggestions for Authors

Reviewer feedback

The research presents a novel and promising hypothesis regarding the role of endometrial stromal senescence in IUA progression, highlighting LGALS9 as a potential therapeutic target. The methodology is sound, and the implications for future treatments are significant, making this a valuable contribution to the field of reproductive medicine. However, further experimental validation and comparison with existing literature are needed to strengthen the claims. Below are some comments and suggestions for the authors’ considerations to enhance the quality of the manuscript.

Comments:

  1. Have the authors performed any immunohistochemistry or qPCR for key proteins (such as LGALS9) to validate findings involving gene expression and immune signaling pathways?

  1. Suggestion to include a more in-depth comparison with existing literature studies on senescence in IUA and related fibrotic conditions. Suggestion to cite related references.

  1. Can the authors offer more insights into the broader cellular mechanisms, including how senescent cells may alter stromal cell behavior and affect embryo implantation? This will provide a more comprehensive understanding of how stromal senescence influences IUA pathophysiology.

  1. Can the authors include a brief comparison of the endometrial senescence data with well-established models of fibrosis in other organs, such as the lung, liver, or kidney in the discussion section of the manuscript. This would help validate whether the molecular pathways identified in IUA (e.g., SASP components, LGALS9) are shared across fibrosis models and if there are tissue-specific variations in senescence. Such comparisons could increase the generalizability of the findings and suggest broader implications for therapeutic strategies.

  1. While the authors perform a cell-type specific analysis of senescence in the endometrial stroma, it would be beneficial to further dissect the role of different stromal cell subtypes (e.g., myofibroblasts, fibroblasts, and mesenchymal stem cells) in the senescence process. Analyzing how these specific subtypes contribute differently to IUA pathology could provide more nuanced insights. Additionally, the study could consider single-cell RNA sequencing in a more detailed manner to uncover any heterogeneity within the stromal cells and their interactions with other cell types during IUA progression.

Author Response

Response to Reviewer 1 Comments

1. Summary

We would like to thank the Reviewer for highlighting the strengths of our study as well as for pointing out several limitations. Please find the detailed responses below and the corresponding revisions/corrections highlighted/in track changes in the re-submitted files.

2. Questions for General Evaluation

Reviewer’s Evaluation

Response and Revisions

Does the introduction provide sufficient background and include all relevant references?

Can be improved

We extended the introduction

Are all the cited references relevant to the research?

Yes

Is the research design appropriate?

Yes

Are the methods adequately described?

Yes

Are the results clearly presented?

Yes

Are the conclusions supported by the results?

Can be improved

We provided extra references to support our conclusions

3. Point-by-point response to Comments and Suggestions for Authors

Comment 1: Have the authors performed any immunohistochemistry or qPCR for key proteins (such as LGALS9) to validate findings involving gene expression and immune signaling pathways?

Response 1: Thanks for the comment. We acknowledge that our results are based solely on bioinformatics and agree that further experimental validation could strengthen our claims. However, experimental validation requires isolation of dozens of endometrial stromal cell lines from patients with IUA. We have encountered several significant limitations regarding IUA patients. Firstly, patients of reproductive age with severe IUA are relatively rare; we had contacted our partner reproductive clinics in Saint-Petersburg and were able to obtain only one cell line suitable for experimental examination in the past year. Secondly, in this study we compared endometrial tissues obtained from healthy endometrium and endometrial tissue from IUA patients that was not obliterated by fibrotic adhesive bands. Most patients with severe IUA possess only small fragments of undamaged endometrium. Thus, to preserve the possibility for reproductive success, clinicians often cannot collect a sufficient amount of tissue necessary for isolating endometrial stromal cells without risking harm to the remaining endometrium. Unfortunately, these circumstances limited our ability to perform experimental validation of our findings in this study. To address this challenge, we validated our results using several independent single-cell and bulk RNA sequencing datasets. Given the lack of appropriate treatment options for IUA and the existence of research laboratories associated with large reproductive centers worldwide that already have or can easily obtain cell lines from IUA patients, we decided to present our findings as they are. We believe that our findings may deepen the understanding of the IUA pathogenesis and facilitate the development of new treatment approaches.

While preparing responses to the Reviewers for our study, we encountered a novel article published a month ago that emphasized the crucial role of endothelial cell senescence in IUA pathogenesis. We paid particular attention to the immunofluorescent co-staining of p21/p16 (senescent cell markers) and CD31 (endothelial cell marker) conducted in this study, as it provides experimental evidence that senescence in the IUA endometrium is not confined to endothelial cells but is likely also present in stromal cells. Although the authors did not address this finding, it underscores the importance of studying stromal senescence in the IUA. Furthermore, it supports our findings regarding the potential role of senescent stromal cells in the pathogenesis of IUA. This information has been included in the Discussion section and is provided in response to comment 2.

Comment 2: Suggestion to include a more in-depth comparison with existing literature studies on senescence in IUA and related fibrotic conditions. Suggestion to cite related references.

Response 2: Thank you for the suggestion. We have added indicated information and relevant references into the discussion section.

In the current version this point is limited to: “Cellular senescence has emerged as a key mediator in organ-specific fibrosis, including idiopathic pulmonary fibrosis, acute kidney injury, and chronic kidney disease, etc. [3,5,6,8–10,30]. In this study, we tested investigated whether enhanced senescence contributes to the development and progression of another fibrotic condition – IUA.”

The updated version: “Cellular senescence has emerged as a key mediator in organ-specific fibrosis, in-cluding idiopathic pulmonary fibrosis, acute kidney injury, and chronic kidney disease, etc [7,9,10,12–14,34]. For instance, biospecimens from the lungs of patients with idio-pathic pulmonary fibrosis exhibited elevated levels of senescence biomarkers p16 and γH2A.X compared to those from healthy controls [9]. Furthermore, senescent cells have been identified in various tissue compartments during chronic kidney disease, including the proximal tubular epithelium, glomeruli, and endothelium [35]. In the context of cardiac fibrosis, the accumulation and persistence of senescent cardiac fi-broblasts contribute to chronic inflammation, collagen deposition, myocardial hyper-trophy, and the progression of fibrosis [36]. Current knowledge indicates that pro-fibrotic effects of senescent cells are primarily mediated by SASP, which comprises pro-inflammatory cytokines such as IL-6, IL-8, CXCL8, CXCL12, TNF-α, MCP-1, MIP-1; reactive oxygen species; matrix metalloproteases; extracellular matrix proteins and remodelers [7,9,10, 12–14, 34–36]. In this study, we investigated whether enhanced stromal senescence contributes to the development and progression of another fibrotic condition – IUA. The only reference to the role of senescence in the IUA progression coincided with our investigation [37]. The authors highlighted the crucial role of en-dothelial cell senescence in IUA pathogenesis, contributing to vascular reduction and fibrosis. Notably, the immunofluorescent co-staining of p21/p16 (senescent cell mark-ers) and CD31 (endothelial cell marker) conducted in this study provides experimental evidence that senescence in the IUA endometrium is not confined to endothelial cells, but is likely also present in stromal cells. Although the authors did not address this finding, it underscores the critical importance of studying stromal senescence in the IUA. Furthermore, it supports our hypothesis regarding the potential role of senescent stromal cells in the pathogenesis of IUA.”

Comment 3: Can the authors offer more insights into the broader cellular mechanisms, including how senescent cells may alter stromal cell behavior and affect embryo implantation? This will provide a more comprehensive understanding of how stromal senescence influences IUA pathophysiology.

Response 3: Thank you for the suggestion. In the previous version of the Discussion section, we highlighted several outcomes of stromal cell senescence for endometrial functioning. To address the suggestion we slightly extended some parts.  

1. The first one concerns the proper formation of a new functional layer during proliferative phase:

“The potential consequences of elevated senescence on tissue functioning stem directly from the primary characteristics of senescent cells: irreversible proliferation arrest and altered auto-, paracrine activity. Regarding the former, the increased senescence observed in the IUA stroma during the proliferative phase may adversely affect EnSC proliferation and the proper formation of a new functional layer. Consistent with this, the endometrium in women with IUA is significantly thinner than that of women without IUA, and EnSC isolated from IUA patients demonstrate reduced colony-forming ability compared to those from healthy women [14,35,36]”.

2. The second one addresses the altered secretory activity of senescent EnSC. This section has been expanded to include additional insights into the possible role of SASP components beyond LGALS9 for the functioning of endometrium. 

Previous version: “To address the functional outcomes of altered secretory activity of senescent cells, we compared the cell-cell communications of normal and IUA stroma. Our findings revealed that stromal cells from IUA patients are “much more sociable” than those from normal tissue, interacting with a broader range of cell types in endometrium. In addition to classical SASP components, stromal cells from IUA patients exhibited a significant increase in LGALS9 expression, thereby creating a pro-inflammatory, immunosuppressive, and profibrotic microenvironment”.

Updated version: “To address the functional outcomes of the altered secretory activity of senescent cells, we compared the cell-cell communications between normal and IUA stroma. Our findings revealed that stromal cells from patients with IUA are “much more sociable” than those from normal tissue, interacting with a broader range of cell types in endo-metrium. Firstly, we detected that EnSC from IUA patients’ displayed enhanced ex-pression of several canonical SASP components. We have previously shown that se-nescent EnSC via SASP can induce paracrine senescence in normal bystander cells, thereby facilitating negative effects of senescence [22]. Furthermore, such inflammato-ry molecules produced by senescent EnSC as CXCL8, CXCL12 may be a source of chronic inflammation in IUA endometrium. Additionally, we revealed that senescent stromal cells in IUA endometrium are characterized by enhanced expression of the urokinase-type plasminogen activator (PLAU), which activated the urokinase-type plasminogen receptor (PLAUR) in myeloid, endothelial, epithelial, stromal, and myo-fibroblast cells. PLAUR is now recognized as a central immune regulating receptor, which can alter intracellular signaling, cell activation, and extracellular adhesion and migration [44]. In addition to classical SASP components, stromal cells from IUA pa-tients exhibited a marked increase in LGALS9 expression, thereby creating a pro-inflammatory, immunosuppressive, and pro-fibrotic microenvironment”.

3. The final point addresses the potential role of EnSC senescence in embryo implantation. Our recent study thoroughly examined the impact of EnSC senescence on embryo implantation [20]. In the updated version, we additionally emphasized this aspect.

Previous version: “Indeed, employing our original approach and generating additional senescence signatures suitable for the decidualized EnSC state, we observed that elevated senescence in the IUA stroma persisted from the proliferative to the mid-secretory phase. According to our previous in vitro findings, senescent EnSC are refractory to decidual stimuli and are unable to give rise either to mature or to senescent decidual cells [20]. The reduced emergence of decidual subpopulations contributes to embryo implantation failure [27–29].”

Updated version: “Indeed, by employing our original approach and generating additional senescence signatures suitable for the decidualized EnSC state, we observed that elevated senes-cence in the IUA stroma persisted from the proliferative to the mid-secretory phase. In our previous study, we conducted a detailed investigation into how senescent EnSC impair embryo implantation [24]. According to our previous in vitro findings, senes-cent EnSC are refractory to decidual stimuli and are unable to differentiate into either mature or senescent decidual cells [24]. The reduced emergence of decidual subpopu-lations is thought to contribute to embryo implantation failure [31–33].”

Comment 4: Can the authors include a brief comparison of the endometrial senescence data with well-established models of fibrosis in other organs, such as the lung, liver, or kidney in the discussion section of the manuscript. This would help validate whether the molecular pathways identified in IUA (e.g., SASP components, LGALS9) are shared across fibrosis models and if there are tissue-specific variations in senescence. Such comparisons could increase the generalizability of the findings and suggest broader implications for therapeutic strategies.

Response 4: Thank you for the suggestion. We had already included description of well-established models of fibrosis in other organs, such as the lung, liver, or kidney in the discussion section of the manuscript, in response to comment 2.

Furthermore, we had already cited recent study that described a novel immunosuppressive subtype of senescent tumor cells that exerted an inhibitory effect on immune cells, including mast cells, myeloid cells, NK cells, B cells, and T cells via LGALS9 signaling [37].

Comment 5: While the authors perform a cell-type specific analysis of senescence in the endometrial stroma, it would be beneficial to further dissect the role of different stromal cell subtypes (e.g., myofibroblasts, fibroblasts, and mesenchymal stem cells) in the senescence process. Analyzing how these specific subtypes contribute differently to IUA pathology could provide more nuanced insights. Additionally, the study could consider single-cell RNA sequencing in a more detailed manner to uncover any heterogeneity within the stromal cells and their interactions with other cell types during IUA progression.

Response 5: Thanks for the comment. In the present study, we initially focused on endometrial stromal cells, as our in vitro senescence signatures are specifically relevant to this cell type. Following integration and cell type annotation, we subsetted stromal cell cluster, excluding myofibroblasts and proliferating stromal cells (Fig. 2A-D). Further analysis of senescence levels conducted solely for stromal cells from healthy and IUA endometrium. To further validate our findings, we performed additional analysis in accordance with the Reviewer’s suggestion. The results of this analysis are presented in Supplementary Figure 2. To begin with, we selected all cells of mesenchymal origin, including stromal cells, myofibroblasts, and proliferating stromal cells, and reintegrated the data to identify any new stromal subclusters that might emerged in IUA and to reveal potential subcluster of senescent cells (Supplementary Figure 2A-D). As shown in Supplementary Figure 2, no cluster specific to IUA was identified (Supplementary Figure 2B) and no distinct senescent cell cluster was observed (Supplementary Figure 2C). Notably, to estimate senescence in all mesenchymal cell types, we applied the SenMayo geneset, as our newly generated signatures are specific to stromal cells. Senescence signal was uniformly distributed throughout the entire cell population within each subtype. However, a pairwise comparison showed that the senescence signal among mesenchymal cell subtypes primarily developed in the stroma (Supplementary Figure 2D). The results of this analysis justify the relevancy of studying stromal cell senescence in IUA pathogenesis.

We added new Supplementary Figure 2 into the manuscript. Also, we added the information provided in this response to the appropriate place in the text.

New Figure legend: Figure S2: Among mesenchymal lineage cell subtypes only stromal cells of patients with IUA exhibit an increased level of senescence during the proliferative phase compared to healthy donors. Normal – healthy donors, IUA – patients with intrauterine adhesions. Analysis was performed on scRNA-seq datasets PRJNA730360 (Normal n = 3) and PRJNA784021 (IUA n = 3), integrated and annotated as demonstrated at Figure 2. (A – D) UMAP representation of mesenchymal lineage cells colored by samples origin, phenotype, cell subtypes, and UCell enrichment scores for the Saul SenMayo geneset, respectively. (E) Violin plots of UCell enrichment scores for the Saul SenMayo geneset for the cell subtypes between Normal donors and patients with IUA.

Previous version: “Indeed, the scores obtained for the “Top 100 up-regulated DEGs” and the “Top 100 down-regulated DEGs” gene signatures confirmed enhanced senescence in thin en-dometrial stroma and exhibited a similar trend tendency as those observed in IUA stroma (Supplementary Figure S1). Collectively, our findings revealed novel genesets for identifying senescence in endometrial stroma, based on the knowledge transferred from the in vitro EnSC senescence program to in vivo tissue samples”.

Updated version: “Indeed, the scores obtained for the “Top 100 up-regulated DEGs” and the “Top 100 down-regulated DEGs” gene signatures confirmed enhanced senescence in thin endometrial stroma and exhibited a similar trend to those observed in IUA stroma (Supplementary Figure S1). To further validate our findings, we performed additional analysis. The results of this analysis are presented in Supplementary Figure 2. To begin with, we selected all cells of mesenchymal origin, including stromal cells, myofibroblasts, and proliferating stromal cells, and reintegrated the data to identify any new stromal subclusters that might emerged in IUA and to reveal potential subcluster of senescent cells (Supplementary Figure 2A-D). As shown in Supplementary Figure 2, no cluster specific to IUA was identified (Supplementary Figure 2B) and no distinct senescent cell cluster was observed (Supplementary Figure 2C). Notably, to estimate senescence in all mesenchymal cell types, we applied the SenMayo geneset, as our newly generated signatures are specific to stromal cells. Senescence signal was uniformly distributed throughout the entire cell population within each subtype. However, a pairwise comparison showed that the senescence signal among mesenchymal cell subtypes primarily developed in the stroma (Supplementary Figure 2D). The results of this analysis justify the relevancy of studying stromal cell senescence in IUA pathogenesis. Collectively, our findings revealed novel genesets for identifying senescence in endometrial stroma, based on knowledge transferred from the in vitro EnSC senescence program to in vivo tissue samples”.

4. Response to Comments on the Quality of English Language

Point 1: The English is fine and does not require any improvement.

Response 1: Thanks

Reviewer 2 Report

Comments and Suggestions for Authors

Comments to the Authors:

This study integrates single-cell and bulk RNA sequencing data to develop endometrial stromal cell (EnSC)-specific senescence gene signatures, revealing significantly elevated senescence levels in the endometrial stromal cells of patients with intrauterine adhesions (IUA) during both the proliferative phase and the secretory phase (window of implantation). The authors propose that LGALS9-targeted immunotherapy may represent a novel intervention strategy for IUA. This work theoretically advances our understanding of stromal senescence in IUA pathogenesis, However, There are some critiques as follows and the paper is not suitable for publication with the current version.

Major points:

  1. The functional roles of LGALS9 are solely supported by bioinformatics analyses. Experimental validation, such as in vitro or in vivo studies, is required to establish causality.
  2. The proposed LGALS9 immunotherapy lacks critical details, including antibody selection, delivery methods, and safety profiles. Preclinical data supporting feasibility and efficacy are essential.
  3. Other senescence-associated factors or pathways are not thoroughly investigated, potentially overlooking key regulatory networks.
  4. The data processing workflow is inadequately described. Enhanced methodological detail is necessary to ensure reproducibility.

Author Response

Response to Reviewer 2 Comments

1. Summary

2. Questions for General Evaluation

Reviewer’s Evaluation

Response and Revisions

Does the introduction provide sufficient background and include all relevant references?

Can be improved

We extended the introduction

Are all the cited references relevant to the research?

Can be improved

We added more references relevant to the study

Is the research design appropriate?

Can be improved

We added explanation of the chosen research design in conclusions section

Are the methods adequately described?

Can be improved

We extended description of the methods

Are the results clearly presented?

Can be improved

We believe that the results are clearly presented in the four main figures. Additionally, we provided three Supplementary tables and two figures

Are the conclusions supported by the results?

Can be improved

We provided extra references to support our conclusions

3. Point-by-point response to Comments and Suggestions for Authors

Comment 1: The functional roles of LGALS9 are solely supported by bioinformatics analyses. Experimental validation, such as in vitro or in vivo studies, is required to establish causality.

Response 1: Thanks for the comment. We acknowledge that our results are based solely on bioinformatics and agree that further experimental validation could strengthen our claims. However, experimental validation requires isolation of dozens of endometrial stromal cell lines from patients with IUA. We have encountered several significant limitations regarding IUA patients. Firstly, patients of reproductive age with severe IUA are rare; we had contacted our partner reproductive clinics in Saint-Petersburg and were able to obtain only one cell line suitable for experimental examination in the past year. Secondly, in this study we compared endometrial tissues obtained from healthy endometrium and endometrial tissue from IUA patients that was not obliterated by fibrotic adhesive bands. Most patients with severe IUA possess only small fragments of undamaged endometrium. Thus, to preserve the possibility for reproductive success, clinicians often cannot collect a sufficient amount of tissue necessary for isolating endometrial stromal cells without risking harm to the remaining endometrium. Unfortunately, these circumstances limited our ability to perform experimental validation of our findings in this study. To address this challenge, we validated our results using several independent single-cell and bulk RNA sequencing datasets. Given the lack of appropriate treatment options for IUA and the existence of research laboratories associated with major reproductive centers worldwide that already have or can easily obtain cell lines from IUA patients, we decided to present our findings as they are. We believe that our findings may deepen the understanding of the IUA pathogenesis and facilitate the development of new treatment approaches.

While preparing responses to the Reviewers for our study, we encountered a novel article published a month ago that emphasized the crucial role of endothelial cell senescence in IUA pathogenesis. We paid particular attention to the immunofluorescent co-staining of p21/p16 (senescent cell markers) and CD31 (endothelial cell marker) conducted in this study, as it provides experimental evidence that senescence in the IUA endometrium is not confined to endothelial cells but is likely also present in stromal cells. Although the authors did not address this finding, it underscores the critical importance of studying stromal senescence in the IUA. Furthermore, it supports our findings regarding the potential role of senescent stromal cells in the pathogenesis of IUA. This information has been included in the Discussion section and is provided.

Comment 2: The proposed LGALS9 immunotherapy lacks critical details, including antibody selection, delivery methods, and safety profiles. Preclinical data supporting feasibility and efficacy are essential.

Response 2: In the present study, we did not aim to develop LGALS9 immunotherapy. We identify that the IUA pathogenesis is associated with the elevated stromal senescence. Based on our bioinformatics findings, along with the existing literature, we just propose LGALS9 immunotherapy as a potential treatment for IUA. We believe that the technical details of LGALS9 immunotherapy are not pertinent to this study. Furthermore, some of the addressed details can be found in the cited references.  

Comment 3: Other senescence-associated factors or pathways are not thoroughly investigated, potentially overlooking key regulatory networks.

Response 3: Thank you for the comment. We have incorporated a discussion of other SASP factors into the relevant sections of the text. However, we believe that the most effective way to minimize the complex profibrotic and proinflammatory effects of senescent cells is through their elimination. In this context, focusing on the specific SASP factors does not appear to be a promising strategy for reducing the overall impact of SASP. An alternative approach is to utilize senolytics specifically designed for the targeted elimination of senescent cells. However, most existing senolytics lack specificity and have significant side effects. In this regard, LGALS9 immunotherapy, which provide the option to enhance the efficacy of natural NK cells-mediated immune clearance of senescent cells, should be considerably safer. We have added this explanation to the text.      

Previous version: “We propose that targeting senescent stromal cells could represent a novel therapeutic approach in the treatment of IUA. Given that traditional senolytic strategies often have side effects due to imperfect selectivity, a more promising alternative for curing IUA may be the LGALS9 immunotherapy protocol, specifically designed to neutralize LGALS9 immunosuppressive activity of senescent cells. This approach may facilitate the restoration of proper immune clearance of senescent cells in the IUA stroma. Either alone or in combination with existing treatment modalities, an LGALS9-based strategy has the potential to effectively address severe IUA”.

Updated version: “We propose that targeting senescent stromal cells could represent a novel therapeutic approach in the treatment of IUA. We believe that the most effective way to minimize the complex pro-fibrotic and pro-inflammatory effects of senescent cells is through their elimination. In this context, focusing on the specific SASP factors does not appear to be a promising strategy for reducing the overall impact of SASP. Moreover, given that traditional senolytic strategies often have side effects due to imperfect selectivity, a more promising alternative for curing IUA may be an LGALS9 immunotherapy protocol, specifically designed to neutralize LGALS9 immunosuppressive activity of senes-cent cells. This approach may facilitate the restoration of proper immune clearance of senescent cells in the IUA stroma. Either alone or in combination with existing treat-ment modalities, an LGALS9-based strategy has the potential to effectively address severe IUA”.

Comment 4: The data processing workflow is inadequately described. Enhanced methodological detail is necessary to ensure reproducibility.

Response 4: Thanks for the comment. We have thoroughly reviewed the description of the operations performed and added minor details that may be helpful for reproducing the analysis.   

4. Response to Comments on the Quality of English Language

Point 1: The English is fine and does not require any improvement.

Response 1: Thanks

Reviewer 3 Report

Comments and Suggestions for Authors

I read with great interest the manuscript titled “Endometrial stromal senescence mediates the progression of intrauterine adhesions” (ID: ijms-3579774), which falls within the aim of International Journal of Molecular Sciences. This study aims to explore the role of endometrial stromal cell senescence in the pathogenesis of IUA, providing original transcriptomic gene signatures to track senescence in both in vitro and in vivo models, and identifying LGALS9 as a potentially targetable immunosuppressive factor.

In my opinion, the manuscript is well-structured and addresses a clinically relevant issue. The authors should be commended for their integration of single-cell and bulk RNA-seq datasets and their effort to develop specific gene sets reflective of stromal senescence. However, to improve the manuscript, the authors are encouraged to address the following points:

  • While the manuscript provides a good rationale for developing EnSC-specific senescence signatures, it would be helpful to further emphasize how these improve upon existing tools such as SenMayo, SnG, or SenCid. What exactly does their approach add in terms of sensitivity, specificity, or tissue relevance?
  • LGALS9 is presented as a central finding, but the link between stromal senescence and LGALS9-mediated immune evasion would benefit from a deeper mechanistic discussion. Additional references on galectin signaling in reproductive tissues and its role in fibrosis/immunosuppression would be valuable.
  • The idea that senescence persists during the secretory phase is compelling. However, it would be useful to expand on how this persistent senescence interferes with decidualization on a functional level, perhaps by discussing known decidual markers or downstream hormonal signaling.
  • The introduction would benefit from a more detailed and updated description of intrauterine adhesions , particularly regarding their clinical manifestations, diagnostic approaches, and pathophysiological mechanisms. It is also recommended to cite the most recent classification systems ( PMID: 40143976; PMID: 38013507, PMID: 39535837).

Author Response

Response to Reviewer 3 Comments

1. Summary

We would like to thank the Reviewer for highlighting the strengths of our study as well as for pointing out several limitations. Please find the detailed responses below and the corresponding revisions/corrections highlighted/in track changes in the re-submitted files.

2. Questions for General Evaluation

Reviewer’s Evaluation

Response and Revisions

Does the introduction provide sufficient background and include all relevant references?

Can be improved

We extended the introduction

Are all the cited references relevant to the research?

Yes

Is the research design appropriate?

Yes

Are the methods adequately described?

Yes

Are the results clearly presented?

Yes

Are the conclusions supported by the results?

Can be improved

We provided extra references to support our conclusions

3. Point-by-point response to Comments and Suggestions for Authors

Comment 1: While the manuscript provides a good rationale for developing EnSC-specific senescence signatures, it would be helpful to further emphasize how these improve upon existing tools such as SenMayo, SnG, or SenCid. What exactly does their approach add in terms of sensitivity, specificity, or tissue relevance?

Response 1: Thanks for the comment. In the present study, we compared our newly generated senescence signatures with two most common SnG (senescence gene) signatures – SenMayo and Fridman senescence up. As shown in Fig. 1D, our Top-up signature distinguishes senescent from non-senescent cells with nearly the same reliability as SenMayo and Fridman. However, only the Top-up signature is specific to stromal cells, while SenMayo and Fridman exhibit high enrichment scores in other cell types (e.g. myeloid, myofibroblasts) (Fig. 1E). This information is already provided in the text: “As shown in Figure 2E, “Fridman senescence up” geneset was found to be more specific to myofibroblasts, while the “Saul SenMayo” demonstrated comparable specificity across most cell types in the endometrium (Figure 2E). Consequently, both genesets are unsuitable for investigating stromal senescence in bulk data. In contrast, the “Top 100 up-regulated DEGs” and the “Top 100 down-regulated DEGs” gene signatures exhibited greater specificity for stromal cells compared to other cell types in the endometrium, with the exception of myeloid cells (Figure 2E). To account for cell composition in the analyzed bulk samples, we applied deconvolution and revealed a significantly higher stromal content in all endometrial samples, while myeloid cells constituted only about 2-3% (Figure 2F). This suggests a minimal, if any, impact of myeloid cells on the assessment of senescence. Consistent with the results obtained from scRNA-seq data, we observed significant upregulation of the Top-up genes and down-regulation of the Top-down genes in IUA compared versus normal endometrium (Figure 2G). These findings support the conclusion of increased stromal senescence in the IUA endometrium.”

SenCid was originally developed based on in vitro stress-induced senescence models. According to the most recent data, SenCid has a good performance for in vitro datasets, but perform poorly in in vivo datasets [https://pmc.ncbi.nlm.nih.gov/articles/PMC10690237/]. This indicates that this tool is unsuitable for whole-tissue RNA-sequencing samples.

To sum up, the best way to study senescence of the desired cell type within tissues is to generate cell type-specific senescence signatures, instead of using universal SnG signatures or machine learning models. This is exactly what we have accomplished.

Previous version: “Currently, numerous genesets and approaches have been described to identify senescent cells in transcriptomic data, including SenMayo, SnG signatures, SenePy, and SenCid [23,31–34]. Unfortunately, none of these approaches appears to be universally applicable. In the present study, we developed an original approach to identify senescence in the endometrial stroma, drawing on knowledge learned from the in vitro EnSC senescence program and applying them to in vivo tissue samples”.

Updated version: “Currently, numerous genesets and approaches have been described to identify senes-cent cells in transcriptomic data, including SenMayo, SnG signatures, SenePy, and SenCid [27,38–41]. Unfortunately, none of these approaches appears to be universally applicable. We believe that the best way to study senescence of the desired cell type within tissues is to generate cell type-specific senescence signatures, instead of using universal SnG signatures or machine learning models. In the present study, we devel-oped an original approach to identify senescence in the endometrial stroma, drawing on knowledge learned from the in vitro EnSC senescence program and transferring this knowledge to in vivo tissue samples”.       

Comment 2: LGALS9 is presented as a central finding, but the link between stromal senescence and LGALS9-mediated immune evasion would benefit from a deeper mechanistic discussion. Additional references on galectin signaling in reproductive tissues and its role in fibrosis/immunosuppression would be valuable.

Response 2: Thanks for the comment. We added missing information into the text.

Previous version: “Currently, LGALS9 has emerged as a promising new target for cancer immunotherapy [38]. Furthermore, it has been established that LGALS9 secreted from EnSC interacts with HAVCR2, an inhibitory receptor expressed on a subset of uterine natural killer (uNK) cells [39].”

Updated version: “Currently, LGALS9 has emerged as a promising new target for cancer immunotherapy [46]. Beyond cancer, it has been observed that LGALS9 plays a role in bone mar-row-derived mesenchymal stem cells-mediated immunosuppression [47]. This occurs by LGALS9 binding to its receptor, HAVCR2, expressed on activated lymphocytes, leading to apoptotic cell death of these activated lymphocytes [47]. Furthermore, the immunosuppressive action of LGALS9 has also been demonstrated in relation to re-productive tissues [48]. Research has shown that the LGALS9/HAVCR2 mechanism is crucial for establishing and maintaining maternal–fetal immune tolerance, which is necessary for maternal immune system to tolerate the semi-allogeneic fetus. Specifi-cally, the authors revealed that trophoblasts induced transformation of peripheral blood natural killer (NK) cells into more immune-tolerant uterine NK (uNK) cells (positive for HAVCR2) through the secretion of LGALS9 and the interaction between LGALS9 and HAVCR2. Additionally, it has been established that EnSC also secrete LGALS9, which may interact with HAVCR2 expressed on a subset of uNK cells [49].”

Comment 3: The idea that senescence persists during the secretory phase is compelling. However, it would be useful to expand on how this persistent senescence interferes with decidualization on a functional level, perhaps by discussing known decidual markers or downstream hormonal signaling.

Response 3: Thank you for the suggestion. Our recent study thoroughly examined the impact of EnSC senescence on embryo implantation [20]. In the updated version, we additionally emphasized this aspect.

Previous version: “Indeed, employing our original approach and generating additional senescence signatures suitable for the decidualized EnSC state, we observed that elevated senescence in the IUA stroma persisted from the proliferative to the mid-secretory phase. According to our previous in vitro findings, senescent EnSC are refractory to decidual stimuli and are unable to give rise either to mature or to senescent decidual cells [20]. The reduced emergence of decidual subpopulations contributes to embryo implantation failure [27–29].”

Updated version: “Indeed, by employing our original approach and generating additional senescence signatures suitable for the decidualized EnSC state, we observed that elevated senes-cence in the IUA stroma persisted from the proliferative to the mid-secretory phase. In our previous study, we conducted a detailed investigation into how senescent EnSC impair embryo implantation [24]. According to our previous in vitro findings, senes-cent EnSC are refractory to decidual stimuli and are unable to differentiate into either mature or senescent decidual cells [24]. The reduced emergence of decidual subpopu-lations is thought to contribute to embryo implantation failure [31–33].”

Comment 4: The introduction would benefit from a more detailed and updated description of intrauterine adhesions, particularly regarding their clinical manifestations, diagnostic approaches, and pathophysiological mechanisms. It is also recommended to cite the most recent classification systems (PMID: 40143976; PMID: 38013507, PMID: 39535837).

Response 4: Thanks for the comment. We expanded Introduction section and added missing references.

Previous version: “Intrauterine adhesions (IUA), also known as Asherman’s syndrome, are characterized by endometrial fibrosis that leads to the partial or complete obliteration of the uterine cavity due to adhesions of the uterine wall [1]. This condition can result in menstrual abnormalities, pelvic pain, infertility, recurrent miscarriages, and abnormal placentation [1]. The majority of IUA cases arise from uterine injuries caused by curet-tage or cesarean sections, while a smaller number are associated with severe endome-trial infections [1]. Currently, hysteroscopic adhesiolysis, postoperative placement of intrauterine contraceptive devices, low-dose aspirin, and regular estrogen therapy are employed as clinical treatments for IUA [1,2]. These standard approaches provide benefits for patients but exhibit a high recurrence rate of endometrial fibrosis [2]. Consequently, numerous novel therapies targeting the pathophysiological mechanisms of IUA are currently under investigation [1]. While some of these emerging approaches show promising results, most – including stem cell therapy (utilizing intact, genetically modified, or preconditioned stem cells) and direct administration of cytokines – address the consequences rather than the underlying causes of the disease. Therefore, a comprehensive investigation of the pathological changes in the endometrium of IUA patients is essential to uncover the molecular mechanisms underlying this condition, which may serve as primary therapeutic targets for effectively treating IUA”.

Updated version: “Intrauterine adhesions (IUA), also known as Asherman’s syndrome, are charac-terized by endometrial fibrosis that leads to the partial or complete obliteration of the uterine cavity due to adhesions of the uterine wall [1]. The main cause of IUA is trau-ma to the endometrial basal layer of the endometrium, mainly post-dilation and curet-tage [2]. However, this condition can also occur after hysteroscopic surgical proce-dures, including hysteroscopic myomectomy and hysteroscopic adenomyomectomy, which breach the basal layer of the endometrium or the myometrium [3]. Common clinical manifestations of IUA are menstrual abnormalities, pelvic pain, infertility, re-current miscarriages, and abnormal placentation [1]. The prognosis and treatment outcomes for IUA patients are closely related to the severity of the disease [4]. Therefore, novel classification systems for IUA are constantly being developed, including those based on comprehensive analysis of the ultrasonographic endometrial thickness, menstrual patterns, reproductive history, and hysteroscopic findings [3–5]. Currently, hysteroscopic adhesiolysis, postoperative placement of intrauterine contraceptive de-vices, low-dose aspirin, and regular estrogen therapy are employed as clinical treat-ments for IUA [1,6]. These standard approaches provide benefits for patients but ex-hibit a high recurrence rate of endometrial fibrosis [6]. Consequently, numerous novel therapies targeting the pathophysiological mechanisms of IUA are currently under in-vestigation [1]. While some of these emerging approaches show promising results, most – including stem cell therapy (utilizing intact, genetically modified, or precondi-tioned stem cells) and direct administration of cytokines – address the consequences rather than the underlying causes of the disease. Therefore, a comprehensive investi-gation of the pathological changes in the endometrium of IUA patients is essential to uncover the molecular mechanisms underlying this condition, which may serve as primary therapeutic targets for effectively treating IUA”.

4. Response to Comments on the Quality of English Language

Point 1: The English could be improved to more clearly express research.

Response 1: Thanks  for the suggestion. We have edited the language of the review with the use of Proofreading & Editing service (https://www.scribbr.com/).

Round 2

Reviewer 2 Report

Comments and Suggestions for Authors

Comments to the Authors:

By applying gene signature, this article found that the senescence level of interstitial cells in endometrium of patients with uterine adhesion increased during the proliferation period, and senescent interstitial cells created a microenvironment of immunosuppression and fibrosis promotion through high expression of LGALS9. Therefore,this paper concluded that interstitial cell senescence can be regarded as the main cause of reduced endometrial reactivity, decreased receptivity and decreased thickness in IUA patients. An LGALS9 immunotherapy regimen specifically designed to neutralize the immunosuppressive activity of senescent cells LGALS9 may offer a promising opportunity to restore effective immune clearance of these cells in the IUA interstitium. However, There are some critiques as follows and the paper is not suitable for publication with the current version.

Major points:

  1. The sample size of some datasets is relatively small. For example, in the single-cell RNA sequencing datasets PRJNA730360 and PRJNA784021, there are only three samples per group, which may limit the statistical power and generalizability of the results. It is suggested that the authors mention in the discussion section the potential impact of sample size on the research results and emphasize the need for future studies to expand the sample size for further validation.
  2. In the discussion section, the interpretation of some key results could be more in-depth. For example, regarding the mechanism of action of LGALS9 in IUA, in addition to immunosuppression and profibrosis, are there other pathways through which it may affect endometrial function? Moreover, the dynamic changes of senescent cells in different menstrual cycle phases and their long-term impact on endometrial repair and regeneration could be discussed more comprehensively.
  3. Although the article is based on a large amount of bioinformatics analysis, it lacks experimental validation. For example, functional validation of the newly generated gene signatures in in vitro or in vivo models, as well as preliminary experimental exploration of the LGALS9 immunotherapy strategy, could be considered to enhance the credibility of the research conclusions.

Author Response

We would like thank the Reviewer for the additional comments and suggestions. We believe that the corrections made substantially improved our study. Below we attach a point-by-point response to the comments of the Reviewer.

Major points:

  1. The sample size of some datasets is relatively small. For example, in the single-cell RNA sequencing datasets PRJNA730360 and PRJNA784021, there are only three samples per group, which may limit the statistical power and generalizability of the results. It is suggested that the authors mention in the discussion section the potential impact of sample size on the research results and emphasize the need for future studies to expand the sample size for further validation.

Thanks for the suggestion. We completely agree with your point. Indeed, the only publicly available single cell RNA sequiencing datasets for IUA are PRJNA730360 and PRJNA784021, which contained relatively small number of samples. However, we additionally validated our results using two independent bulk RNA-seq datasets: (1) GSE224093, which includes 7 healthy and 7 IUA samples, (2) PRJNA916532, which includes 4 healthy and 5 IUA samples. The results are presented in Fig. 1G. As recommended by the Reviewer, we included a limitations section in the Discussion to emphasize the need for future studies.

Updated version: “The results of the present study are based solely on bioinformatic analysis; therefore, several important limitations should be highlighted. Firstly, the sample size of the publicly available scRNA-seq datasets analyzed in this study is relatively small, indicating a clear need for future research to expand the sample size for further validation. Secondly, the study lacks experimental validation. Consequently, functional validation of the newly generated gene signatures in in vitro or in vivo models, as well as preliminary experimental exploration of the LGALS9 immunotherapy strategy, should be further considered to enhance the credibility of the research conclusions”.     

  1. In the discussion section, the interpretation of some key results could be more in-depth. For example, regarding the mechanism of action of LGALS9 in IUA, in addition to immunosuppression and profibrosis, are there other pathways through which it may affect endometrial function? Moreover, the dynamic changes of senescent cells in different menstrual cycle phases and their long-term impact on endometrial repair and regeneration could be discussed more comprehensively.

Thank you for the suggestion. The detailed investigation and discussion of the senescence reaction across menstrual cycle phases is provided in our previous study, which is cited in the text [24]. Here we investigated possible impact of heightened stromal senescence in proliferative phase that resulted in reduced emergence of senescent and mature decidual cells that emerged during secretory phase. Both subpopulations are required for implantation, insufficient emergence of both subpopulations during IUA may result in reduced endometrial receptivity and further infertility. We can only speculate that persistence of increased amounts of senescent cells during menstrual phase may negatively affect scarless healing of endometrium. This speculation was added in the Discussion section. Furthermore, we added new data on LGALS9 and decidualization published on April 2025 [Expression of co-signaling molecules TIM-3/Galectin-9 at the maternal-fetal interface].

Previous version: “Consistent with our in vitro findings, we demonstrated that heightened senescence in the IUA stroma during the WOI resulted in a decreased emergence of both decidual subpopulations and predecidual cells. Interestingly, IUA is typically characterized by a hormonally unresponsive endometrium that is unable to support proper embryo implantation [52].”

Updated version: “Consistent with our in vitro findings, we demonstrated that heightened senescence in the IUA stroma during the WOI resulted in a decreased emergence of predecidual cells as well as both mature and senescent decidual subpopulations that are required for proper embryo implantation. Interestingly, IUA is typically characterized by a hormonally unresponsive endometrium that is unable to support proper embryo implantation [52]. In cycles without conception, heightened stromal senescence may affect “scarless” healing of endometrium. Although certain amounts of senescent cells are essential for wound healing by favoring the plasticity of different cell populations necessary for wound closure, incomplete elimination and accumulation of these senescent cells could promote an impaired tissue formation due to the deposition of collagen, leading to a fibrotic tissue [53]”.

Previous version: “This approach may facilitate the restoration of proper immune clearance of senescent cells in the IUA stroma. Either alone or in combination with existing treatment modal-ities, an LGALS9-based strategy has the potential to effectively address severe IUA.”

Updated version: “This approach may facilitate the restoration of proper immune clearance of senescent cells in the IUA stroma. Either alone or in combination with existing treatment modalities, an LGALS9-based strategy has the potential to effectively address severe IUA. At the same time, according to the most recent data LGALS9 expression in stromal cells is closely linked to their degree of decidualization [54]. Therefore, the potential application of LGALS9 immunotherapy for IUA curing should be strictly limited to proliferative phase of the menstrual cycle, in order to avoid any undesirable impact on decidualization.”

  1. Although the article is based on a large amount of bioinformatics analysis, it lacks experimental validation. For example, functional validation of the newly generated gene signatures in in vitro or in vivo models, as well as preliminary experimental exploration of the LGALS9 immunotherapy strategy, could be considered to enhance the credibility of the research conclusions.

Thanks for the comment. This comment reiterates your previous suggestion that we have already addressed in the first round:

“Response 1: Thanks for the comment. We acknowledge that our results are based solely on bioinformatics and agree that further experimental validation could strengthen our claims. However, experimental validation requires isolation of dozens of endometrial stromal cell lines from patients with IUA. We have encountered several significant limitations regarding IUA patients. Firstly, patients of reproductive age with severe IUA are rare; we had contacted our partner reproductive clinics in Saint-Petersburg and were able to obtain only one cell line suitable for experimental examination in the past year. Secondly, in this study we compared endometrial tissues obtained from healthy endometrium and endometrial tissue from IUA patients that was not obliterated by fibrotic adhesive bands. Most patients with severe IUA possess only small fragments of undamaged endometrium. Thus, to preserve the possibility for reproductive success, clinicians often cannot collect a sufficient amount of tissue necessary for isolating endometrial stromal cells without risking harm to the remaining endometrium. Unfortunately, these circumstances limited our ability to perform experimental validation of our findings in this study. To address this challenge, we validated our results using several independent single-cell and bulk RNA sequencing datasets. Given the lack of appropriate treatment options for IUA and the existence of research laboratories associated with major reproductive centers worldwide that already have or can easily obtain cell lines from IUA patients, we decided to present our findings as they are. We believe that our findings may deepen the understanding of the IUA pathogenesis and facilitate the development of new treatment approaches.

While preparing responses to the Reviewers for our study, we encountered a novel article published a month ago that emphasized the crucial role of endothelial cell senescence in IUA pathogenesis. We paid particular attention to the immunofluorescent co-staining of p21/p16 (senescent cell markers) and CD31 (endothelial cell marker) conducted in this study, as it provides experimental evidence that senescence in the IUA endometrium is not confined to endothelial cells but is likely also present in stromal cells. Although the authors did not address this finding, it underscores the critical importance of studying stromal senescence in the IUA. Furthermore, it supports our findings regarding the potential role of senescent stromal cells in the pathogenesis of IUA. This information has been included in the Discussion section and is provided.”

According to the provided response, we are currently unable to conduct functional in vitro or in vivo validation due to insufficient cell or tissue material. We have also included this point in the limitations section:

Updated version: “The results of the present study are based solely on bioinformatic analysis; therefore, several important limitations should be highlighted. Firstly, the sample size of the publicly available scRNA-seq datasets analyzed in this study is relatively small, indicating a clear need for future research to expand the sample size for further validation. Secondly, the study lacks experimental validation. Consequently, functional validation of the newly generated gene signatures in in vitro or in vivo models, as well as preliminary experimental exploration of the LGALS9 immunotherapy strategy, should be further considered to enhance the credibility of the research conclusions”.

Round 3

Reviewer 2 Report

Comments and Suggestions for Authors

Comments to the Authors:

To a certain extent, you responded to the previous review comments, especially making efforts in the discussion of sample size limitations, result verification, and in-depth interpretation of key results. However, there are still some key issues in the research, especially in the aspects of experimental verification and the exploration of some mechanisms. These problems may affect the reliability of the research conclusions and the application prospects. It is not recommended to publish for the time being.

Major points:

  1. Although the author mentioned in the reply the difficulty of obtaining samples from IUA patients, the lack of experimental verification remains a major flaw of this study. Although bioinformatics analysis can provide valuable clues, it cannot replace experimental verification. For instance, the functional verification of newly generated gene signatures in in vitro or in vivo models, as well as the preliminary experimental exploration of the LGALS9 immunotherapy strategy, are crucial for enhancing the credibility of research conclusions.The authors are encouraged to conduct experimental validation whenever possible. For example, they could use existing cell lines or animal models to functionally validate the newly generated gene signatures or conduct preliminary explorations of the LGALS9 immunotherapy strategy. If obtaining IUA patient samples remains challenging, collaborating with other research institutions to share sample resources could be a solution.
  2. Although the author further discussed the mechanism of action of LGALS9, the discussion on the dynamic changes of senescent cells at different menstrual cycle stages and their long-term effects on endometrial repair and regeneration is still not comprehensive enough. For instance, apart from LGALS9, are there any other signaling pathways involved in the impact of senescent cells on endometrial function? Are there any interactions among these pathways?The authors should delve deeper into the mechanisms by which senescent cells affect endometrial function, especially their dynamic changes across different menstrual cycle phases and interactions with other cells and signaling pathways. This may require combining various experimental techniques, such as co-culture of cells, gene knockout, or overexpression.

Author Response

Reviewer 2 comments

Thanks for the comments. We believe that the manuscript was revised substantially to address the comments of all the Reviewers. In our opinion, there is no need for another Review round; since the points raised are the same as in previous rounds and we have nothing to add. We hope that final decision can be made by the Editors based on the revision history.    

Major points:

  1. Although the author mentioned in the reply the difficulty of obtaining samples from IUA patients, the lack of experimental verification remains a major flaw of this study. Although bioinformatics analysis can provide valuable clues, it cannot replace experimental verification. For instance, the functional verification of newly generated gene signatures in in vitro or in vivo models, as well as the preliminary experimental exploration of the LGALS9 immunotherapy strategy, are crucial for enhancing the credibility of research conclusions.The authors are encouraged to conduct experimental validation whenever possible. For example, they could use existing cell lines or animal models to functionally validate the newly generated gene signatures or conduct preliminary explorations of the LGALS9 immunotherapy strategy. If obtaining IUA patient samples remains challenging, collaborating with other research institutions to share sample resources could be a solution.

Newly generated signatures were generated based on the RNA-sequencing data obtained for our EnSC lines (control and senescent), so there is no need to verify these signatures using EnSC cell lines. As we indicated in the study and in our previous responses, these results were previously published [24]. Even though we are unable to experimentally verify our bioniformatic data just now (as we indicated several times), we believe that the analysis provided is valid (we confirmed our results using four independent single cell and bulk RNA-sequencing datasets) and bring novelty to the field.   

  1. Although the author further discussed the mechanism of action of LGALS9, the discussion on the dynamic changes of senescent cells at different menstrual cycle stages and their long-term effects on endometrial repair and regeneration is still not comprehensive enough. For instance, apart from LGALS9, are there any other signaling pathways involved in the impact of senescent cells on endometrial function? Are there any interactions among these pathways?The authors should delve deeper into the mechanisms by which senescent cells affect endometrial function, especially their dynamic changes across different menstrual cycle phases and interactions with other cells and signaling pathways. This may require combining various experimental techniques, such as co-culture of cells, gene knockout, or overexpression.

We have previously extended discussion section on this topic, according to your suggestion in the round 2. Further description is redundant, since it is irrelevant to the study.

Round 4

Reviewer 2 Report

Comments and Suggestions for Authors

Comments to the Authors:

By applying gene signature, this article found that the senescence level of interstitial cells in endometrium of patients with uterine adhesion increased during the proliferation period, and senescent interstitial cells created a microenvironment of immunosuppression and fibrosis promotion through high expression of LGALS9. Therefore,this paper concluded that interstitial cell senescence can be regarded as the main cause of reduced endometrial reactivity, decreased receptivity and decreased thickness in IUA patients. An LGALS9 immunotherapy regimen specifically designed to neutralize the immunosuppressive activity of senescent cells LGALS9 may offer a promising opportunity to restore effective immune clearance of these cells in the IUA interstitium. The manuscript meets the publication criteria and is recommended for acceptance.